# *Campylobacter jejuni* motility integrates specialized cell shape, flagellar filament, and motor, to coordinate action of its opposed flagella

Eli J. Cohen[1][☯], Daisuke Nakane[2][☯], Yoshiki Kabata[2], David R. Hendrixson[3], Takayuki Nishizaka[2], Morgan Beeby[1] *

1 Department of Life Sciences, Imperial College London, London, United Kingdom, 2 Department of Physics, Gakushuin University, Tokyo, Japan, 3 Department of Microbiology, University of Texas Southwestern Medical Center, Dallas, Texas, United States of America

☯ These authors contributed equally to this work.
* m.beeby@imperial.ac.uk

**Data Availability Statement:** All relevant data are within the manuscript and its Supporting Information files.

## Abstract

*Campylobacter jejuni* rotates a flagellum at each pole to swim through the viscous mucosa of its hosts' gastrointestinal tracts. Despite their importance for host colonization, however, how *C. jejuni* coordinates rotation of these two opposing flagella is unclear. As well as their polar placement, *C. jejuni's* flagella deviate from the norm of Enterobacteriaceae in other ways: their flagellar motors produce much higher torque and their flagellar filament is made of two different zones of two different flagellins. To understand how *C. jejuni's* opposed motors coordinate, and what contribution these factors play in *C. jejuni* motility, we developed strains with flagella that could be fluorescently labeled, and observed them by high-speed video microscopy. We found that *C. jejuni* coordinates its dual flagella by wrapping the leading filament around the cell body during swimming in high-viscosity media and that its differentiated flagellar filament and helical body have evolved to facilitate this wrapped-mode swimming.

## Author summary

*Campylobacter jejuni* is a leading cause of gastroenteritis worldwide. This species uses its helical body and opposing flagella to drill its way through the viscous mucosa of host organisms' gastrointestinal tracts. In this work, we show that *C. jejuni* coordinates its two opposing flagella by wrapping the leading flagellum around the cell body when swimming in viscous environments. We also provide evidence that the helical cell body of *C. jejuni* and its composite flagellar filament are important for wrapping and unwrapping of the flagellar filament during reversals of swimming direction.

**Funding:** This work was supported by Medical Research Council grant MR/P019374/1 to MB, NIH grant R01AI065539 to DRH, and in part by Japan Society for the Promotion of Science KAKENHI Grants JP16H06230 to DN and JP15H04364 to TN, and by the Precise Measurement Technology Promotion Foundation to DN. The funders had no role in study design, data collection and analysis, decision to publish, or preparation of the manuscript.

**Competing interests:** The authors have declared that no competing interests exist.

# Introduction

Motility is a potent selective benefit for microbes, and understanding motility mechanisms is central to understanding microbial cell biology and for development of novel antibiotics. Many Bacteria swim using flagella, rotary helical propellers spun by ion-flux-driven rotary motors embedded in the cell envelope. While the motility of model organisms is well understood, how flagella are used for propulsion differs substantially in other Bacteria. The well-studied *E. coli* and *Salmonella enterica* coordinate randomly-oriented flagellar motors by bundling their flagella into a helical bundle using a universal joint—the "hook"—at the base of the flagellar filament that redirects rotating filaments into a bundle [1–3]. Many other species, however, have polar motors, necessitating mechanisms to avoid the opposing action of motors from opposing poles. *Caulobacter*, *Shewanella*, and some *Helicobacter* species leave one pole unflagellated [4–6]; *Magnetospirillum magneticum* inversely coordinates its opposing motors [7]; and spirochaetes assemble motors of different composition at opposing poles [8], suggesting that they also have a mechanism for inverse coordination. It is unknown, however, how other polar flagellates coordinate their opposing motors.

*Campylobacter jejuni* is a polar flagellate that causes several million cases of gastroenteritis annually [9,10] and requires its flagella for gut colonization [11–13]. *C. jejuni* constructs one flagellum at each pole, a pattern known as amphitrichous flagellation, to swim at speeds approaching 100 μm/second, and characteristically swims faster in moderately viscous fluids than it does at low viscosities [14]. Other factors contributing to *C. jejuni*'s swimming ability include a higher-torque flagellar motor, facilitating motility through viscous environments such as the protective, viscous layer of mucous lining the digestive tract; a helical cell shape important for colonization of hosts that may enable drilling through viscous mucous; and a flagellar filament composed of two distinct regions that are constructed from two nearly identical flagellin monomers. Together, these unique aspects of *C. jejuni*'s flagellation produce darting motility involving repetitive short runs and reversals, although how the two flagella coordinate to facilitate these reversals remains unclear.

Here we asked how *C. jejuni* swims productively despite its two apparently opposed flagella. We constructed mutants of *C. jejuni* 81–176 whose flagella we could fluorescently label, enabling us to visualize flagellar movement in different environments and genetic backgrounds. We found that both filaments are always left handed, that *C. jejuni* wraps one flagellum around its cell body, and that the cell body is a right-handed helix. Direction changes involve reversal of polarity of wrapping, so that the previously wrapped filament unfurls, while the previously unwrapped filament wraps around the cell body. Our results also demonstrate that the two regions of the flagellar filament are important for wrapping and that helical cell shape plays a role in efficient filament unwrapping during reversal of swimming direction.

# Results

## *C. jejuni* coordinates its opposing motors by wrapping the leading filament around its cell body in high-viscosity media

To observe flagellar filaments during swimming, we changed the serine at position 397 of the *flaA* gene to a cysteine to allow for labelling with maleimide-conjugated fluorophores (*e.g.* DyLight-488 maleimide). *flaA* encodes the major flagellin, which constitutes the majority of the flagellar filament. We selected S397 because it is glycosylated and therefore likely surface-exposed, and mutation to an alanine has little effect on motility [15]. We found that our *flaA*^S397C mutant (EJC28) exhibited wild type motility in both soft-agar swim assays and liquid culture. Labeled and unlabeled cells swam at comparable speeds in liquid culture,

 

demonstrating that labeling of filaments and cell bodies of EJC28 did not affect swimming behavior (**S1 Fig**). Throughout the remainder of this manuscript, "WT" refers to EJC28.

After labelling the flagellar filaments and cell bodies, WT cells were suspended in Mueller-Hinton (MH) broth and observed by high speed video. We directly observed rotation of filaments during swimming in liquid culture. Under our standard conditions flagella rotated at approximately 100 Hz, comparable to *E. coli* [2], when monitored at 1600 frames/s (**S1 Movie**). All subsequent observations of swimming behavior were collected at 400 frames/s, sufficient to capture a rotation rate of 100 Hz.

When cells were observed swimming in MH broth, about half (55%) of cells had both filaments projecting outward from their respective poles. We were surprised, however, to find that the remaining 45% of cells had one filament wrapped around the cell body during swimming. In cells with a wrapped filament, the leading filament wrapped around the cell body, while the lagging (i.e., pushing) flagellum remained unwrapped (**Fig 1A**, **S2 Movie**).

We sought to understand how this wrapping behavior might contribute to *C. jejuni*'s motility in a higher-viscosity medium similar to its native environment. To mimic this environment, we supplemented MH broth with 0.15–1.0% methylcellulose to give solutions with viscosities ranging from ~25–150 mPa·s (25–150 cP), a physiologically relevant range that would be encountered by a cell of *C. jejuni* as it navigates through the gastrointestinal tract [16,17]. Methylcellulose solutions are loose networks of long-chain polymers that echo the molecular architecture of mucous [18], and behave differently at the microscale to homogenous Newtonian viscous liquids such as Ficoll solutions.

We found that increasing the concentration of methylcellulose increased the proportion of wrapped cells. In broth without methylcellulose added, about 50% of the cells we observed exhibited wrapped-mode swimming. At ≥0.3% methylcellulose, almost all of the cells (≥95%) exhibited wrapped-mode swimming (**Fig 1B**, **S2 Movie**).

The increase in wrapping at higher viscosity was accompanied by an increase in swimming velocity (**Fig 1C and 1D**). Increased velocity at higher viscosities has been observed in a number of flagellated species with helical cell shapes [18]. While the swimming velocity of both wrapped and unwrapped cells were approximately equal in MH broth lacking methylcellulose (~25 μm/second), in 0.15% methylcellulose, wrapped cells swam significantly faster than unwrapped cells (44 μm/second vs. 30 μm/second, respectively. **Fig 1E**). We conclude that efficient wrapping is a function of viscosity and that wrapping facilitates faster swimming.

## Two motors are better than one: the wrapped flagellum contributes to propulsion

We wondered whether wrapped flagella contribute to cell propulsion or whether wrapping is a mechanism for parking an otherwise opposing flagellum, coiled and inactive around the cell body. To ascertain whether wrapped flagella rotate, we suspended cells in MH broth with 25% w/v Ficoll, a Newtonian fluid that we found slows cell body rotation. Wrapped filaments still rotated around the slowed cell body in these conditions, showing that wrapped filaments are active rather than being held immobilized against the cell body (**S3 Movie**).

To determine how much wrapped flagella contribute to propulsion, we compared the swimming velocity of singly-flagellated cells with either unwrapped-lagging filaments or wrapped-leading filaments to doubly-flagellated cells in MH + 0.5% methylcellulose. To acquire singly-flagellated cells we capitalized on a Δ*fliI* mutant. FliI is an ATPase from the flagellar type III secretion system that contributes to assembly but is less important for *C. jejuni* than other organisms, and whose deletion does not affect the torque-generating parts of the motor [19]. Deletion of *fliI* renders most cells in the population aflagellate, but the few cells

 

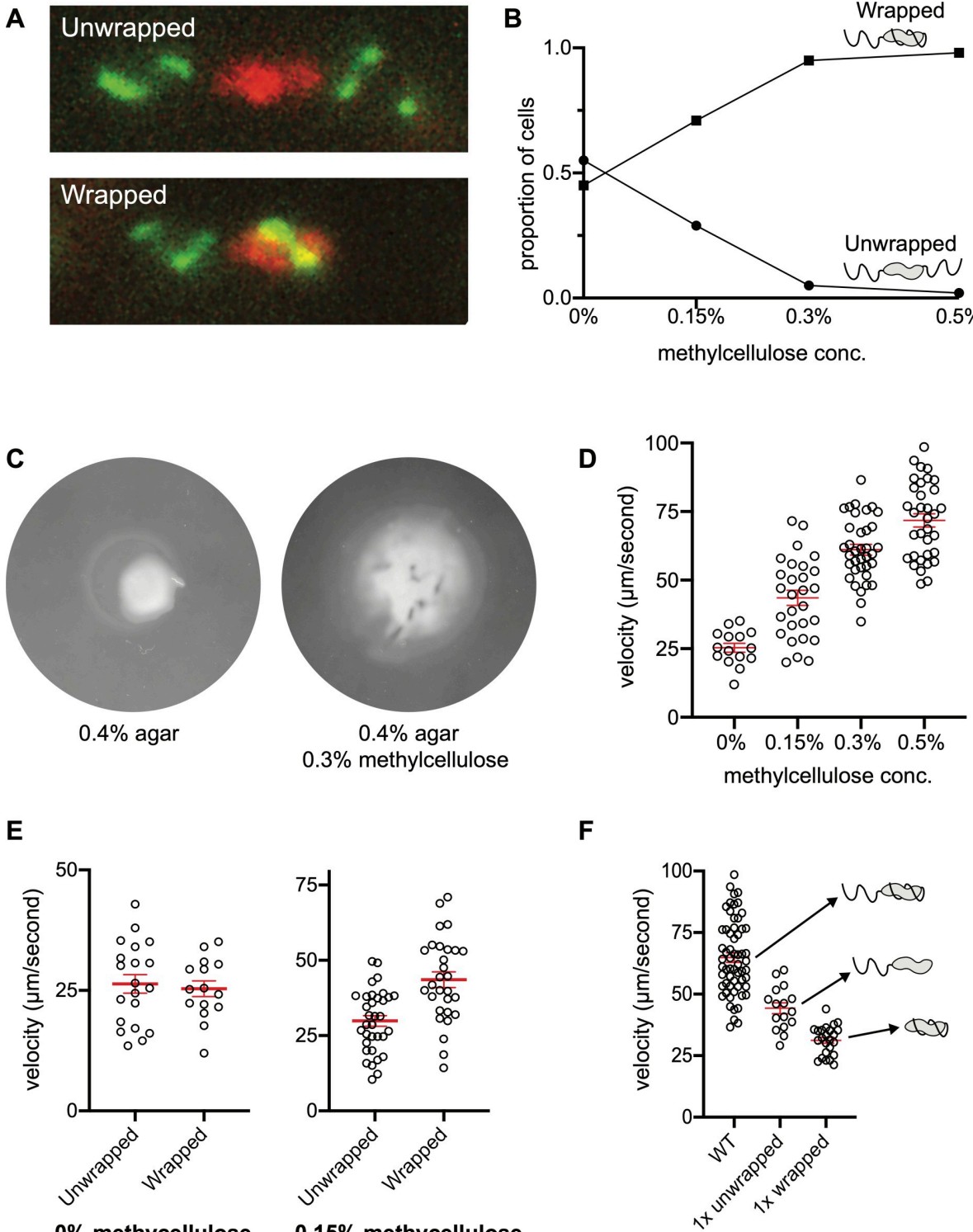

**Fig 1. *Campylobacter jejuni* wraps the leading flagellum and swims faster in viscous media.** As the viscosity of the media increases, the proportion of cells with wrapped leading flagellar filaments also increases (**A** and **B**). In addition to the increase in wrapping, both the swim-halo diameter in soft agar and single-cell swimming velocity increase substantially in the presence of methylcellulose. (**C** and **D**). In MH broth without methylcellulose added, unwrapped and wrapped cells swim at comparable velocities, while wrapped cells outperform unwrapped cells as velocity increased (**E**). Doubly-flagellated wrapped cells swim faster than singly-flagellated (*i.e.* Δ*fliI*) unwrapped or wrapped cells (**F**). In **B**, 100–150 cells in each methylcellulose concentration were used to determine the proportion of wrapped to unwrapped cells in the population. In **D**, **E** and **F**, red bars represent average velocity ± SEM.

that do construct flagella tend to possess only one, of approximately WT length (**S2 Fig**). These filaments polymerize to WT length, and have distributions of FlaA and FlaB indistinguishable from the WT distribution, although we cannot rule out subtle structural or biomechanical differences to WT filaments.

In high-viscosity media, singly-flagellated Δ*fliI C. jejuni* exhibited both unwrapped swimming, with the unwrapped flagellum pushing from behind the cell, and wrapped swimming, with the wrapped flagellum coiling around the cell (**S4 Movie**). Singly-flagellated wrapped cells swam ~40% slower than singly-flagellated unwrapped cells. Doubly-flagellated WT *C. jejuni* with a pushing flagellum augmented with a wrapped flagellum swam faster than uniflagellated Δ*fliI C. jejuni* cells with either a pushing or wrapped flagellum (**Fig 1F**). These results demonstrate that wrapped flagella actively contribute thrust in viscous environments.

## Directional switching involves a polarity switch of wrapped and unwrapped flagella

Switching direction is central to chemotaxis [20,21], but it is unclear how *C. jejuni* cells achieve their characteristic back-and-forth pattern of swimming reversals. Naively, this might be accomplished by reversing the direction of both motors to reverse flagellar thrust, with the unwrapped-lagging filament becoming an unwrapped-leading filament. By labeling flagella, we were able to observe their behavior during swimming reversals. We found that in high-viscosity media, directional switching occurs in three steps. First, the leading filament reverses rotational direction while wrapped around the cell body. Second, after approximately one full revolution, this filament unwraps from the cell body. Third, the initially lagging flagellum wraps around the cell body while the initially wrapped flagellum becomes the lagging, unwrapped flagellum (**Fig 2**, **S5 Movie**). Using Total Internal Reflection Fluorescence Microscopy (TIRFM), we observed that both filaments retained left-handed helices before, during and after directional switching (**S6 Movie**). We infer from this that in both the initial forward, and final "reversed" forward swimming, both motors spin left-handed filaments CCW, and the hook facilitates redirection of torque of the filament that is wrapped around the cell body

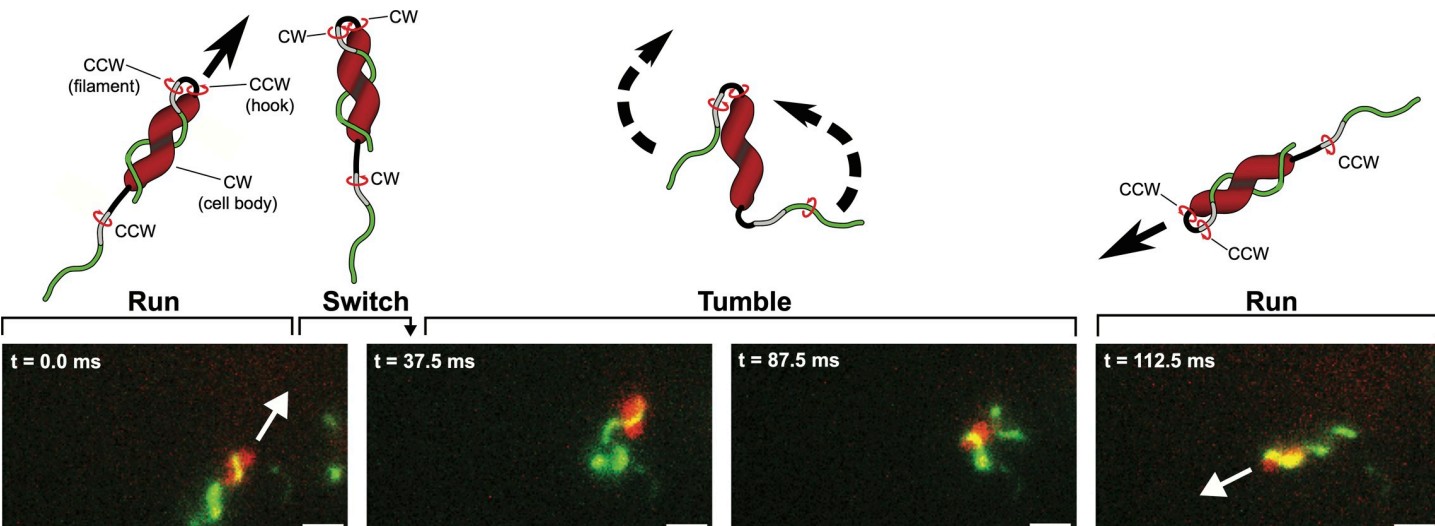

**Fig 2. Changing swimming trajectory involves a switch in wrapped filament polarity.** Wrapped cells change swimming direction, or tumble, by switching which filament is wrapped. The cell comes to a stop when motor rotation switches from CCW to CW, which also causes the wrapped filament to unwrap. The once lagging, unwrapped filament then wraps around the cell body while the previously wrapped filament becomes the trailing, unwrapped filament. Scale bars equal to 1.5 μm.

by 180˚ so that both filaments exert thrust in the same direction. Cell reversal is therefore the result of a polarity change of which filament is wrapped around the cell body and which filament lags behind the cell body, with the initial and final states equivalent except for a switch in which filament is wrapped.

### The cell body has opposite handedness to both flagella, which both spin CCW by default

That cell bodies of helical bacteria contribute thrust for swimming predicts that the helical *C. jejuni* cell body has the opposite handedness of its flagellum, because the motor's counter-rotation of a cell body and filament of the same handedness would produce opposing thrusts. We confirmed this prediction using TIRFM, finding that both wrapped and unwrapped filaments are always left-handed (LH) while the helical cell body is always right-handed (RH) (**Fig 3A and 3B**, **S7 Movie**). TIRFM also revealed that the cell body and filament have comparable helical angles of ~40˚ (**Fig 3C**). During runs, lagging unwrapped flagella rotate CCW while cell bodies counterrotate CW. Both wrapped and unwrapped filaments from the same cell rotate at approximately the same rate to form a coherent, propagating wave.

To confirm that both flagella spin in the same direction, we locked motors in CCW rotation using a *cheY* knockout in the *flaA*^S397C background [22,23]. Cells in this background swam at comparable speeds to WT *C. jejuni*, including wrapped-mode swimming at high viscosity (**S8 Movie**), although they did not make directional switches and had a non-motile phenotype on soft-agar plates due to their inability to chemotax. This result strongly suggests that rotation of both wrapped and unwrapped flagella is CCW.

### Wrapping is driven by motor rotation and drag on the filament

We next asked whether wrapping is driven by motor rotation or extrinsic factors. We noted that our singly-flagellated Δ*fliI C. jejuni* cells wrapped their sole filament, indicating that motor rotation, not cell translation, is required for wrapping: during switching, cells do not move, and the only force exerted on the filament is from motor torque.

We also occasionally observed cells with one paralyzed flagellum, which did not wrap around the cell body (**S9 Movie**), further demonstrating that wrapping is an active process

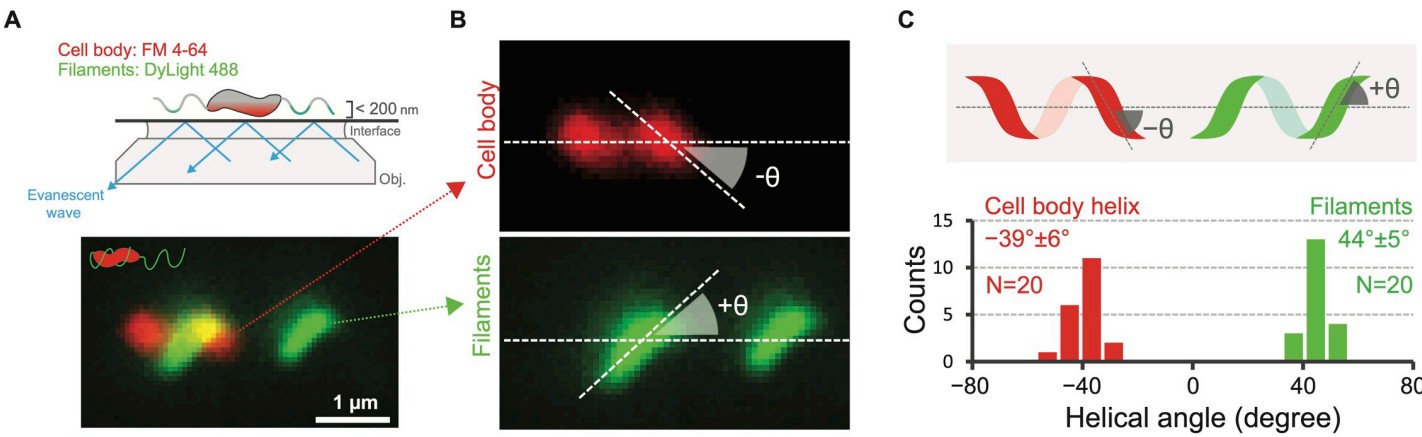

**Fig 3. The cell body and flagellar filament have opposite handedness.** TIRFM revealed that the cell body is a right-handed (RH) helix, while the filament assumes a left-handed (LH) helical conformation (**A** and **B**), and that the filament remains a LH helix whether it is wrapped or unwrapped. The pitch of both the flagellar filament and cell body are approximately identical at ~40˚ (**C**).

requiring a rotating flagellar motor. Whether paralyzed flagella are defective or have a braking or clutching mechanism is unclear.

In addition to motor rotation, efficient filament wrapping also depends on the medium in which the cell is swimming. As viscosity increases, so too does the proportion of cells in the population that have wrapped filaments. Furthermore, the proportion of wrapped cells in the population decreases as swimming velocity decreases; slower cells are less likely to have wrapped leading filaments, although this effect was only seen in mutants with filaments that are more rigid than those of the WT strain (see below). This suggests that drag on the filament increases wrapping efficiency. Taken together, these data show that wrapping requires factors both intrinsic and extrinsic to the cell.

## Helical cell shape is important for efficient unwrapping

Although helical and straight *C. jejuni* cells swim comparably in low viscosity solutions, previous studies have shown that helical cells swim almost twice as fast as straight cells in high-viscosity solutions of long unbranched polymers such as methylcellulose. Previous studies, however, did not record the behavior of individual flagella during swimming. We speculated that slower swimming speed of straight cells in high-viscosity media may be due to an impact of cell shape on wrapping.

To test this hypothesis, we generated a straight-cell mutation in the *flaA*$^{S397C}$ background, allowing us to visualize filament wrapping in non-helical cells. To make cells straight we deleted *pgp1*, which encodes Pgp1 (peptidoglycan peptidase 1), a periplasmic muropeptidase that processes peptidoglycan as a prerequisite for helical cell shape [24].

In high-viscosity media, the Δ*pgp1* mutant exhibited WT-like wrapped swimming and increased swimming velocity. In WT cells, the leading, wrapped filaments almost always unwrap upon switching of swimming direction (~70%). Unlike WT cells, however, we found that few Δ*pgp1* cells were able to unwrap their filaments during directional switching (~25%) (**Fig 4A and 4B**). During switching events with no unwrapping, the wrapped filament switches rotational direction as it tries but fails to peel free from the cell body. When individual Δ*pgp1* cells were tracked over multiple switches, wrapped filaments would only infrequently break free from the cell body, allowing the opposing filament to wrap.

The failure of Δ*pgp1* cells to unwrap means that the CCW-to-CW-to-CCW motor-rotation switching sequence often does not result in the change of wrapped-filament polarity seen for WT cells. Thus, rather than the run-reverse-run WT darting phenotype (*i.e.*, where a reversal is due to a polarity switch of wrapped and unwrapped filaments), many Δ*pgp1* cells instead exhibit a run-pause-run phenotype, with the pause representing a switch in the direction of motor rotation and failure to unwrap for the polarity switch (**S10 Movie**). The two runs, as a result, are in the same direction.

We did also observe cells that reversed swimming direction despite failing to unwrap upon switching, instead reversing with the leading filament pulling the cell, presumably because both filaments rotated CW for an extended time (**S11 Movie**). These cells swam slower than the normal CCW-rotating, unwrapped-lagging/wrapped-leading configuration, consistent with slower rotation of the motor during CW rotation than CCW rotation [25].

## Helical cell shape also contributes to propulsion

We wondered whether cell helicity also contributes thrust or whether the unwrapping defect of the Δ*pgp1* mutant fully explains why straight cells swim slower in high viscosity media. To address this, we measured swimming velocities of individual cells, comparing doubly-

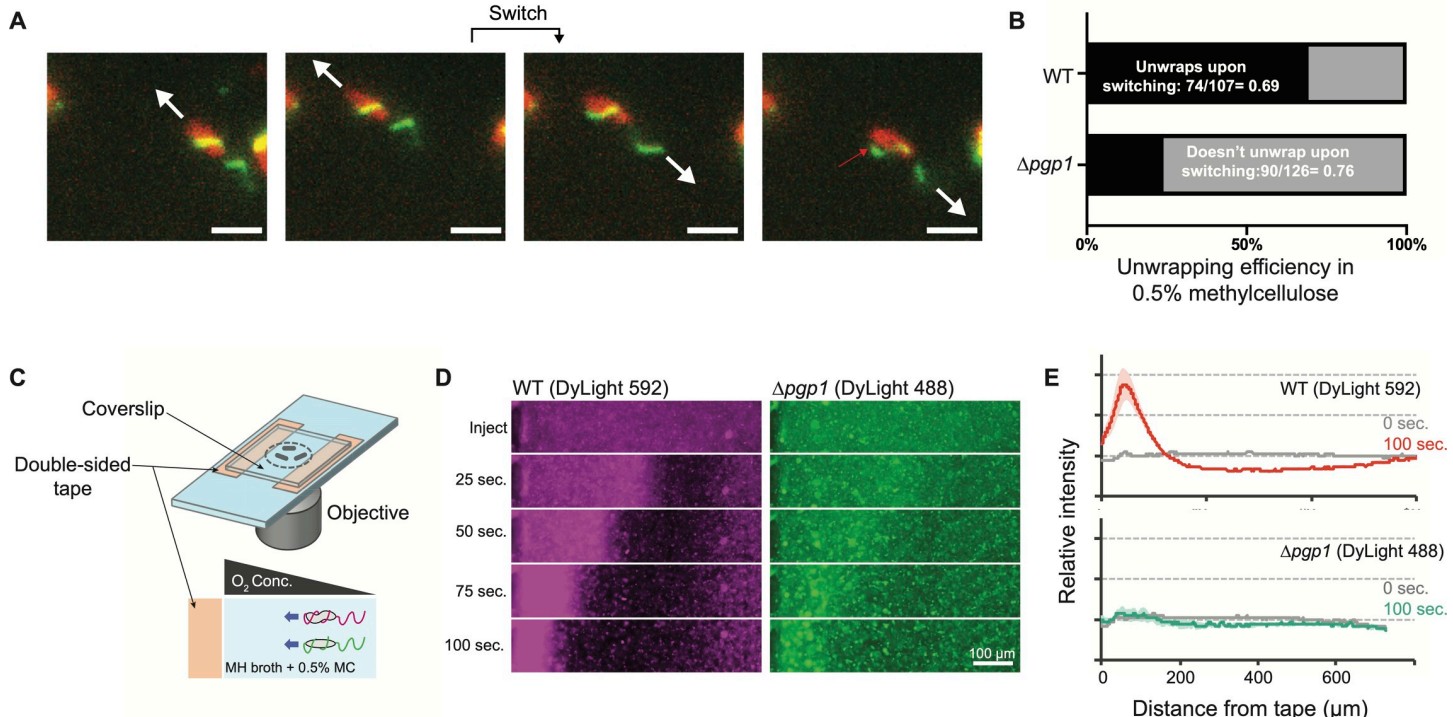

**Fig 4. Unwrapping efficiency and chemotactic ability are impaired in straight cells.** Switching of motor rotation from CCW to CW usually results in filament unwrapping in the WT, but not in straight cells. During reversals, the wrapped filament of Δ*pgp1* cells will often peel away from the cell body, but fail to fully unwrap (red arrow) (**A**). In the straight cell background, filament unwrapping upon switching only occurs 26% of the time, compared to 69% in helical cells (**B**). An aerotaxis competition (**C**) between WT and Δ*pgp1* cells revealed that, in addition to reaching the O₂-rich border (double-sided tape) first, the WT also formed a high density, persisting swarm at the border that left few stragglers behind (**D** and **E**). Scale bars in **A** equal to 2 μm. Fluorescence intensity signals in **E** are the average of three replicate experiments for each mutant.

flagellated WT and Δ*pgp1* cells with leading wrapped flagella and lagging unwrapped flagella to assess differences in linear velocity.

At high viscosity, Δ*pgp1* *C. jejuni* swam at ~75% the speed of WT *C. jejuni*. To more fully isolate the effects of helical cell shape on swimming speed, we generated a singly-flagellated Δ*pgp1* Δ*fliI* mutant and compared the swimming speeds of these cells to our otherwise-WT, singly-flagellated Δ*fliI* strain. In both unwrapped and wrapped conformations, singly-flagellated non-helical cells swam 70–75% the velocity of their corresponding WT counterparts (**S3 Fig**). This demonstrates that helical cell shape does contribute to propulsion, but that that contribution is minor, as also recently found in *H. pylori* [26]. It is also possible that in the *pgp1* knockout the biomechanical properties of the cell body are altered in such a way as to reduce swimming efficiency (*e.g.* decreased stiffness of the cell body), which our data do not allow us to rule out at this time. Nevertheless, there is no evidence for substantial changes in body stiffness in our videos.

## Efficient unwrapping is important for tactic proficiency

We speculated that the impaired ability to reverse swimming direction in the Δ*pgp1* background would impact its tactic ability. To test this, we performed a competition assay between the WT and Δ*pgp1* strains, independently labeling each strain with different maleimide-conjugated fluorophores before combining both strains in the same sample chamber at equivalent concentrations to compare population-level aerotactic behavior of straight and helical cells (**Fig 4C**).

Both WT and Δ*pgp1* cells aerotactically migrated toward the higher concentration of oxygen at the taped edges of the sample chamber (**Fig 4D**, **S12 Movie**, **S13**–**S15 Movies**). Unsurprisingly, the faster-swimming WT cells reached the edge of the sample chamber before the Δ*pgp1* cells. WT cells also, however, formed a higher density swarm than that of the Δ*pgp1* mutant. The WT swarm migrated to the edge of the sample chamber, where oxygen concentration is highest, and persisted there, leaving few "stragglers," *i.e.* the fluorescent signal localizes almost entirely to the edge of the sample chamber. In contrast, the Δ*pgp1* swarm was both less dense than the WT and also left many stragglers (**Fig 4E**). We conclude that efficient unwrapping of the filament is important for taxis, which may partially explain the colonization defect of straight-cell mutants in *in vivo* infection assays [27].

## Two differentiated flagellins facilitate wrapping by making a filament with nuanced mechanical properties

Whereas species such as *Salmonella* and *E. coli* construct their filaments entirely from a single type of flagellin [28], the *C. jejuni* filament is a composite of two different flagellins, FlaA and FlaB [29]. FlaA and FlaB are 95% identical and are the result of a relatively recent duplication event, distinct from the duplication of analogous FlaA and FlaB in *Shewanella* species (**S4 Fig**). In *C. jejuni*, *flaB*, encoding what is known as the minor flagellin, is expressed prior to *flaA* along with other early flagellar structural genes (*e.g.* hook and rod genes) [30–32]. Being expressed before *flaA*, FlaB polymerizes on top of the completed hook before *flaA* transcription initiates, to form the first ~250–1000 nm cell-proximal stub of the filament. The remaining ~3–4 μm of the filament is composed of FlaA [33]. In analogous *Shewanella putrefaciens*, relative positioning and amounts of two flagellins regulates how easily flagella screw around the cell [6].

We sought to assess whether the two flagellins contribute to wrapping. Although a labelable version of *flaB* in a *flaA*$^+$ background mainly produced fluorescent stubs too short for analysis, we occasionally observed cells that constructed full-length filaments with FlaB distributed along their length. Strikingly, we noticed that these filaments tended not to wrap when swimming in viscous media (**S16 Movie**), suggesting to us that FlaB dampens the wrapping tendency of a filament, and may form a filament that is more rigid than a FlaA filament.

To explore the contributions of FlaA and FlaB to filament rigidity, we constructed mutants with filaments composed entirely of FlaA or FlaB. In each case, the flagellin is expressed from the *flaA* (σ$^{28}$) promoter and is terminated by the *flaB* transcriptional terminator (S5 Fig). As above, we observed that cells with filaments composed entirely of FlaB tended not to wrap in high viscosity media, but that this depended on which area of the microscope slide we observed.

We noticed that when we observed cells in the middle of the sample chamber, they swam at much lower velocity than cells close to the double-sided tape of the sample chamber or cells in close proximity to an air bubble trapped between the cover slips. Cells in the oxygen-depleted middle portion of the sample chamber swam at about half the velocity as those near an oxygen source (S6 Fig). We speculate that the difference in velocity arises from reduced proton motive force (PMF), which drives motor rotation, in the oxygen-starved cells. All observations of cells in this work were made in close proximity to the double sided-tape (within ~20 μm) of the sample chamber containing cell suspensions.

In the case of the all-FlaB mutant, cells near the double-sided tape almost always had the leading filament wrapped, whereas the leading filaments of cells in the middle of the sample chamber tended to stay unwrapped (**S17 Movie**). WT and all-FlaA cells in high viscosity media were predominantly wrapped regardless of where they were observed in the sample

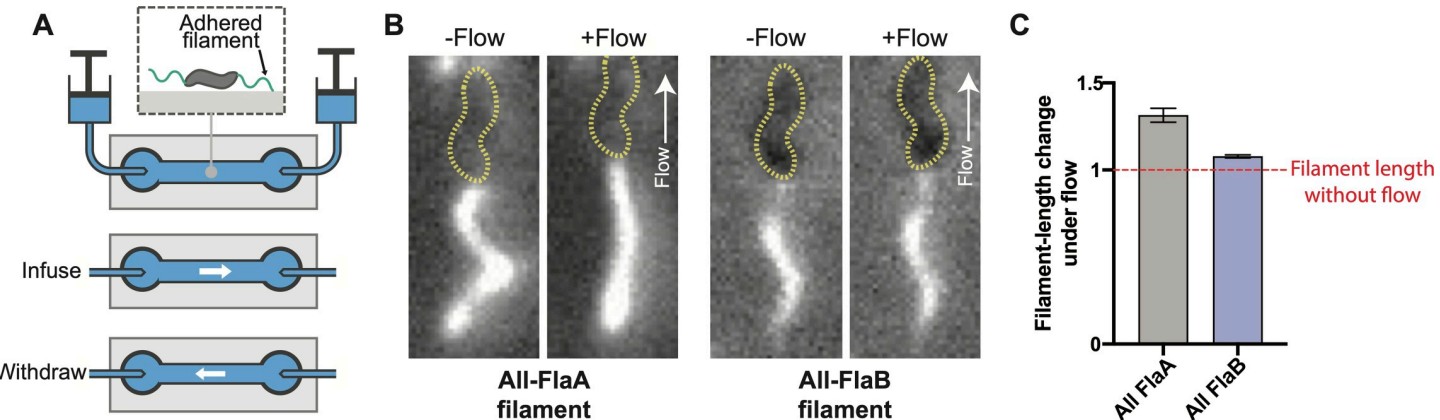

**Fig 5. FlaB forms a more rigid filament than FlaA.** To determine whether the FlaA portion of the filament is less rigid than the FlaB portion, a flow chamber was constructed that allowed us to flow buffer over labeled, immobilized cells (**A**). When buffer was flowed past WT (*i.e. flaA*$^{S397C}$ *flaB*$^+$) or all-FlaA filaments, they stretched to become ~30% longer than their relaxed length. In contrast, all-FlaB filaments stretched only ~10% under the same flow conditions (**B** and **C**). For **C**, 16 all-FlaA filaments and 20 all-FlaB filaments were measured for elongation under flow, error bars represent the SEM for each.

chamber or how fast they were swimming. Since the amount of drag experienced by the leading filament will increase with swimming speed, this result suggests that more force is required to wrap an all-FlaB filament around the cell than either a WT or all-FlaA filament.

If FlaB filaments are more rigid than FlaA filaments, a FlaB filament will resist deformation when force is applied more than a FlaA filament. To test this, we constructed a flow chamber to observe filaments in fluid flows of defined rate (**Fig 5A**). We fluorescently labelled filaments of WT, all-FlaA and all-FlaB cells, paralyzed flagella by addition of CCCP, applied cells to the flow chamber and allowed them to adhere to the coverslip. We then flowed buffer through the chamber at a constant rate. Both WT and all-FlaA filaments stretched by up to 30% when flow was applied, while all-FlaB filaments only stretched up to ~10% under the same flow rate (**Fig 5B and 5C**, **S18** and **S19** Movies). Taken together, our results support a model of the filament where the FlaB basal portion of the filament is more rigid than the cell-distal FlaA portion.

## Discussion

We sought to understand how *Campylobacter jejuni* coordinates its opposed flagella for productive swimming. By fluorescently labeling flagellar filaments, we found that *C. jejuni* coordinates thrust from both filaments in the same direction by wrapping one filament around the cell; that reversals in darting motility are due to a polarity reversal in which filament is wrapped; and that helical cell shape is crucial for unwrapping of filaments during reversals.

To effectively swim with two opposing flagellar motors, *C. jejuni* wraps its leading filament around its cell body while its unwrapped lagging filament pushes from behind. Deletion of *cheY*, locking motors in CCW rotation, does not affect forward motility, indicating that hook rolling transmits torque to spin the wrapped filament CCW, producing unified thrust from two opposing CCW-rotating motors. Although some species wrap filaments around their cell body to escape confined spaces [34] wrapped-mode swimming is essential for *C. jejuni* motility in liquid environments, suggesting that *C. jejuni* has evolved wrapped-mode swimming to enable assembly of two otherwise opposing motors. Indeed, wrapped swimming has also recently been observed in related amphitrichous *Helicobacter suis*, which assembles tufts of 4–6 flagella at each pole and wraps the leading tuft around the cell body when swimming in viscous media [35]. Being able to swim with a motor at each pole has clear advantages: *C. jejuni*

swims faster with two motors than one, and dual flagellation means that both daughter cells inherit a functional motor immediately after division.

Our results also unveil a previously unappreciated significance of the characteristic helical cell shape of *C. jejuni*. It has been presumed that helical cells assist drilling through the long-chain polymers of the viscous mucous of the GI tract [18]. Indeed, our results support this: straight-cell mutants of *C. jejuni* do not penetrate high-viscosity motility agar as readily as the WT (**S7 Fig**). Nevertheless, recent apparently paradoxical discoveries indicate that helical cell shape contributes relatively little to motility [26]. Our results demonstrate another important function: a helical cell body of opposite handedness to the filament facilitates immediate unwrapping of flagellar filaments from the cell body upon reversal; in straight cells, filaments often fail to unwrap. We speculate that the filament has an affinity for the cell surface, and that a right-handed helical cell minimizes contact with a wrapped, left-handed filament. Indeed, in other contexts, attraction between the filament and cell body is selectively beneficial and produces autoagglutination. Autoagglutination depends upon glycosylation of *C. jejuni* flagellin and cell-surface lipooligosaccharide [15,36], and *C. jejuni* mutants with increased autoagglutination swim together as linear chains of up to four cells [37]. Intriguingly, these cell chains do not reverse direction, consistent with inhibition of flagellar unwrapping.

Wrapping and unwrapping result from an interplay between motor rotation, a composite filament with specifically tuned rigidity, and the environment through which the cell swims. Wrapping requires an active motor, as the leading filaments of cells with non-functional motors do not wrap around the cell body, but are instead dragged alongside the cell. Wrapping also requires viscous drag on the filament to promote efficient wrapping; motor rotation itself is not sufficient.

Our data suggest that wrapping also depends upon conformational changes in the flexible FlaA region of the leading filament when it encounters drag over a certain threshold. While in lophotrichous species the wrapping of the flagellum around the cell body requires CCW-to-CW switching, this does not appear to be the case in *C. jejuni*. We observed many cells in lower viscosity regimens in which a leading, unwrapped flagellum spontaneously wraps around the cell body without switching motor rotation. The presence of wrapped cells in the Δ*cheY* background further indicates that CCW-to-CW switching is not required for wrapping to occur. This is supported by the observation that a leading, all-FlaB filament will bend in high viscosity media but fail to wrap unless the cell is swimming at sufficiently high velocity as well as the fact that WT filaments do not wrap efficiently in low viscosity regimens.

That both all-FlaA and all-FlaB mutants swim through both regular and high-viscosity swim agar with similarly reduced efficiencies demonstrates that the composite filament has nuanced, superior biomechanical properties over a single-flagellin filament in *C. jejuni*'s native environment (**S8 Fig**). FlaB may impose an invariant left-handed helical filament regardless of rotational direction, as with other polar flagellates, but unlike peritrichous *E. coli* and *Salmonella*. FlaB constitutes up to the first 1 μm of the filament, and forms a short, curved stub barely sufficient to complete one helical turn. The rigid FlaB stub may transmit protofilament asymmetry from the curved hook to impose helicity on the more flexible FlaA filament during rotation, since an all-FlaA filament has a tendency to be straight, rather than helical (**S20 Movie**) (Inoue et al. 2018). Differences between FlaA and FlaB are primarily in the D0 and D1 regions, which form the stable filament core. In *Salmonella* [38], *Pseudomonas aeruginosa*, and *Bacillus subtilis* [39] mutations in these regions alter the equilibrium of filament handedness, consistent with our observed differences. Molecular structures of both FlaA and FlaB will be needed to better understand these aspects. The inability to form a right-handed helix may partially be because it is not needed during chemotaxis to disrupt a filament bundle.

A recent study in *Shewanella putrefaciens* contributes to understanding duplication of flagellins in polar flagellates. Like *C. jejuni*, *S. putrefaciens* has a composite filament with a more rigid, cell-proximal portion and a less-rigid cell distal portion. In *S. putrefasciens*, a filament composed entirely of the cell-proximal flagellin forms a helix with narrower diameter and greater pitch than the helix formed by the cell distal flagellin. Kühn *et al.* suggest that having a portion of the filament with a tighter helix increases stability at the base of the filament and helps to prevent unnecessary wrapping. Nevertheless, *C. jejuni* flagellin duplication is a parallel convergent evolution with *Shewanella*; *C. jejuni* FlaA and FlaB are more similar to one-another than they are even to other relatively closely-related species such as *H. pylori*, suggesting multiple independent duplications. Correspondingly, the mechanical implications on composite filaments are different: in contrast to *S. putrefasciens*, the filaments of an all-FlaB *C. jejuni* mutant have comparable helical pitches and lengths to all-FlaA filaments, suggesting that a tighter helical coil in the FlaB filament is not responsible for the dampening of wrapping in *C. jejuni*.

Our study has a number of limitations. Our videos do not allow us to determine what occurs during wrapped-to-unwrapped transitions in detail. Specifically, the mechanism by which unwrapping occurs remains unclear. Since the wrapped filament is capable of rotating around the cell body both CCW or CW, it remains to be determined what happens to the hook and/or filament upon CCW-to-CW switching that results in rapid unwrapping of the filament. A recent report on switching and hook dynamics in *Salmonella* demonstrated that upon CCW-to-CW switching, the hook transiently locks such that it no longer rotates around its axial centerline. In *Salmonella*, hook locking serves to drag the filament from the bundle behind the cell toward the front of the cell [40]. We imagine that a similar process might be involved in unwrapping of the filament in *C. jejuni*, whereby hook locking serves to pull the wrapped filament from the cell body upon CCW-to-CW switching.

Although our Δ*cheY* mutant demonstrates that the hook rolls as a universal joint to retain CCW rotation of the axial structures even when bending by 180°, we cannot rule out that some of the motor's rotation also drives partial rotation of the hook around the motor axis. Indeed, it is very likely that the hook also rotates around the motor axis, which would explain why we see filaments spinning around the cell body.

What proportion of motor rotation drives hook rolling and what proportion drives hook rotation remains to be investigated, and it is important to note that we are not directly observing motor rotation. Indeed, with the data presented here, we cannot definitively rule out the possibility that the wrapped motor rotates CW while the unwrapped rotates CCW (*i.e.* asymmetric rotation) Nevertheless, there is a strong precedent for most of motor rotation translating as rolling of the hook based on the behavior of hooks from peritrichous bacteria [41], and the alternative hypothesis–that the hook attached to the wrapped filament rotates around the leading motor axis and does not roll, and the leading motor rotates CW while the lagging motor rotates CCW–would require phosphorylated CheY to somehow exert the opposite effect on two identical motors simultaneously to ensure coordinated switching during reversal of swimming directions.

Indeed, there are clear mechanistic models to explain how specific bacteria inversely coordinate the action of motors at opposing poles, unlike *C. jejuni*, whose motility genes have been exhaustively surveyed [42]. In *Borrelia burgdorferi*, asymmetric motor rotation at opposing poles has been demonstrated, and is explained by structurally distinct motors at opposing poles: motors at both poles incorporate FliG2, while a second, abnormal FliG (FliG1) localizes to only one pole; deletion of *fliG1* causes paralysis of one pole [8,43]. Because *C. jejuni* has only one copy of *fliG* in its genome there is no precedent to think that any such mechanisms facilitate coordinated and opposite actions of motors at the two poles. A recent report on

*Magnetospirillum magneticum*, another amphitrichous species that wraps its leading flagellum, invokes a model for wrapped-mode swimming that involves asymmetric motor rotation [7]. *M. magneticum* has 32 different *cheY*s in its genome and two different sets of *motAB*, which may facilitate coordinated opposite actions of motors at the two poles. *C. jejuni*, in contrast, has only one *cheY* and one set of *motAB*. Since, however, this study relied on data comparable to ours (*i.e.* fluorescently-labeled filaments and high-speed-video microscopy), asymmetric rotation in *M. magneticum* remains to be unambiguously demonstrated.

Finally, it is unclear how unwrapped cells in low-viscosity media are able to swim at all. In MH broth lacking any methylcellulose, unwrapped and wrapped cells, in both the WT and Δ*cheY* backgrounds (**S21 Movie**), swim at approximately the same low velocity. But how two unwrapped filaments don't cancel each other out is a mystery. We speculate that this could be due to the two filaments having slightly different lengths, or possibly due to the leading and lagging flagellum assuming different helical pitches.

Together, our results enable us to propose a model for *C. jejuni* motility. Wrapped-mode swimming enables thrust generation from the two CCW-rotating LH flagella as well as the CW-rotating RH helical cell body. For taxis, the leading, wrapped flagellum swaps to become the lagging, unwrapped flagellum, and vice versa, enabling a dramatic change of direction. Indeed, *C. jejuni*, which inhabits crevices in the gastrointestinal tract, may use it to make 180° reversals to escape confined spaces. The helical cell shape contributes to more than thrust and penetrance, however, and also serves to minimize filament-cell interactions that contribute to autoagglutination. The host colonization defect of straight cell mutants is likely partially due to taxis defects stemming from their inability to reverse. That both the cell-body helix and filament helix have similar 40° helical angles of opposite handedness may be an optimization for both thrust generation and reducing filament-cell contact area.

Our findings also have implications for evolution. We have previously noted how complex adaptations become essential in the evolution of the *C. jejuni* motor. Here, we observed how cell biology adaptations also became essential: in this case, the helical cell body, which may first have evolved to generate thrust, has been exapted to counteract filament-cell interactions, and has thus now become essential for effective motility.

## Materials and Methods

### Bacterial growth conditions

Strains of *C. jejuni* used in this study (**Table 1**) were routinely cultured on MHT agar (Mueller-Hinton agar (Difco) supplemented with 10 μg/mL trimethoprim (Sigma-Aldrich)). When

**Table 1. List of strains used in this study.**

| Strain # | Genotype | Ref. | Notes |
|---|---|---|---|
| **WPK440** | Δ*flaAB::cat-rpsL* pRY108::*flaA*$^{S398C}$ | | from Hendrixson lab |
| **EJC28** | *flaA*$^{S397C}$ | This study | |
| **EJC30** | Δ*fliI flaA*$^{S397C}$ | This study | |
| **EJC31** | Δ*cheY::aphA3'-rpsL flaA*$^{S397C}$ | This study | |
| **EJC33** | Δ*pgp1 flaA*$^{S397C}$ | This study | |
| **EJC35** | Δ*pgp1* Δ*fliI flaA*$^{S397C}$ | This study | |
| **EJC37** | *flaB*$^{S398C}$ | This study | |
| **EJC39** | Δ*flaAB::flaB*$^{S398C}$ | This study | all-FlaB$^{S398C}$ filament |
| **EJC40** | Δ*flaAB::flaB*$^{S397C}$ | This study | all-FlaB$^{S397C}$ filament |
| **EJC41** | Δ*flaAB::flaA*$^{S397C}$ | This study | all-FlaA$^{S397C}$ filament |

necessary, additional antibiotics were added at the following concentrations: kanamycin (Km), 50 μg/mL; chloramphenicol (Cm), 10 μg/mL; streptomycin (Sm), 2 mg/mL.

For strain construction, motility agar assays and negative-stain electron microscopy (performed at Imperial College London), cultures were grown in a Heracell 150i trigas incubator (Thermo-Fischer) set to 5% $O_2$, 10% $CO_2$, 85% $N_2$ at 37˚C.

For fluorescence microscopy experiments (performed at Gakushuin University), cultures were grown at 37˚C using Mitsubishi Anaeropack MicroAero gas generator packs.

## Strain construction

With the exception of strain WPK440, all strains used in this study were constructed via natural transformation of linear splicing-by-overhang-extension (SOE) PCR products (*i.e.* no plasmids were used or constructed during strain construction). The alleles constructed in this study are all chromosomally encoded (with the exception of WPK440).

Construction of all EJC mutants followed the same two-step protocol: an *aphA3'-rpsL* cassette (conferring kanamycin (Km$^R$) resistance and streptomycin sensitivity (Sm$^S$)) bearing ~500 bp homology to the genomic locus to be mutated was generated by SOE PCR. ~5 ug of the SOE PCR product was then transformed into DRH212 (a Sm$^R$ derivative of *C. jejuni* 81–176), or a derivative thereof (*e.g. flaA*$^{S397C}$), via the biphasic method [44] for 4–8 hours under microaerobic conditions. Transformants were then selected on MHT agar supplemented with Km. After 48–72 hours, Km$^R$ colonies were picked and purified, then screened for Sm$^S$.

Replacement of the *aphA3'-rpsL* cassette followed essentially the same protocol: a SOE PCR fragment bearing the desired mutation with ~500 bp flanking homology arms was transformed into the Km$^R$, Sm$^S$ intermediate strain, allowed to incubate for 4–8 hours without selection before plating serial dilutions on MHT agar supplemented with 2 mg/mL Sm. Sm selection plates were incubated for ~24 hours at 37˚C in a 2.5 L culture jar with a Campygen microaerobic-environment-generating gas pack (Oxoid), then transferred to the Heracell 150 incubator for an additional 2–3 days until colonies were large enough to pick. Sm$^R$ colonies were screened for Km$^S$ and Sm$^R$ and then single colony purified. Km$^S$, Sm$^R$ transformants were sent for Sanger sequencing (Source Biosciences) to verify construction of the desired allele.

Strain WPK440 (*C. jejuni* 81–176 *rpsL*$^{Sm}$ *flaAB*::*cat-rpsL*/pRY108::*flaA*$_{S398C}$) was constructed using procedures previously described [32,45]. Briefly, pABT1081 [46] was electroporated into DRH212 and plated to MH with chloramphenicol to obtain ABT1173 (81–176 *rpsL*$^{Sm}$ *flaAB*::*cat-rpsL*). pWPK280 was created by designing primers to amplify DNA fragments encompassing the *flaA* locus from ~200 bp upstream and ~100 bp downstream of the coding sequence along with a point mutation within *flaA* to result in replacement of the codon for S397 with one for cysteine. These PCR fragments were then cloned into pRY108 by Gibson assembly and transformed into DH5α/pRK212.1 [47, 48] and then conjugated into ABT1173 to generate WPK440.

## Motility agar assays

Motility agar experiments were performed by spotting 2 μL of cell suspension, adjusted to an O.D.$_{600}$ of 1, to MHT motility agar (0.4% w/v agar concentration). Suspensions were allowed to dry on the agar surface, followed by incubation at 37˚C for ~24 hours. Swimming proficiency in motility agar was determined by measuring the diameter of the swim halo following incubation.

High-viscosity motility agar was made by adding 0.3% w/v methylcellulose (4,000 cP, Sigma-Aldrich) to MHT motility agar. To do so, a 0.6% w/v methylcellulose solution was autoclaved and then stirred overnight at 4˚C to allow the methylcellulose to dissolve. Following

overnight incubation at 4˚ C, the methylcellulose solution was warmed to ~40˚ C and combined 1:1 with molten 0.8% w/v MHT agar cooled to ~40˚ C and poured before the final 0.4% agar, 0.3% methylcellulose media could solidify. Motility assays were then carried out as described above.

For determining swimming ability in motility agar, five replicates of each strain to be tested were performed. Swimming ability is reported as the mean +/- SEM.

## Labeling of cells for fluorescence microscopy

Flagellar filaments bearing cysteine residues were labeled using DyLight maleimide-conjugated fluorophores (Thermo-Fischer). Cell bodies were counterstained using FM 4–64 lipophilic membrane stain (Life technologies).

For all fluorescence microscopy experiments performed in this study, the same labeling protocol was followed: Frozen glycerol stocks of a mutant of interest were incubated on MHT agar at 37˚ C for 24–48 hours, followed by subculturing to a fresh MHT plate for an additional ~16 hours. Cells were then suspended in 1 mL PBS at an $O.D._{600}$ of ~1.0–1.5, to which 1 μL DyLight dye (10 μg/μL in DMSO) was added. Cell suspensions were incubated with DyLight for ~30 minutes at 37˚ C, after which time 1 μL FM 4–64 (10 μg/μL in DMSO) was added to the suspension and gently mixed. Immediately after addition of FM 4–64, suspensions were washed 2x in PBS and resuspended in MH broth. Cell suspensions with different concentrations of methylcellulose were made by combining labeled-cell suspensions with MH+1% methylcellulose broth in the desired ratio.

## Optical microscopy

For visualization of DyLight 488-labeled flagella, a blue laser beam (OBIS488-20; Coherent) was introduced into an inverted fluorescence microscope (IX71; Olympus) equipped with 100× objective lenses (UPLSAPO 100×OPH, 1.4 N.A.; Olympus), dichroic mirrors (Di01-R488; Semrock), dual-view imaging system (DV2; Photometrics, 565dcxr; Chroma, FF03-535/50 and BLP01-568R; Semrock), a CMOS camera (Zyla 4.2; Andor), and an optical table (HAX-0806; JVI). Projection of the image to the camera was made at 65 nm per pixel. Sequential images of cells were acquired by the imaging software (Solis; Andor) as 16-bit images with a CMOS camera under 2.5-ms or 5-ms intervals for free-swimming under epi-illumination and TIRFM, respectively, and converted into a sequential TIF file without any compression.

For measurements of swimming cells under low magnification, cells were visualized under a phase-contrast microscope (IX83; Olympus) equipped with 20× objective lens (UCPLFLN 20×PH, 0.7 N.A.; Olympus), a high-speed camera system (LRH1540; Digimo) and an optical table (RS-2000; Newport). Projection of the image to the camera was made at 240 nm per pixel. Sequential images of cells were acquired as 8-bit images with a CMOS camera under 10-ms intervals, and converted into a sequential TIF file without any compression.

For flow experiments, labeled flagella were visualized under a fluorescence microscope (IX83; Olympus) equipped with 100× objective lens (UPLSAPO 100×OPH, 1.4 N.A.; Olympus), a filter set (FITC-5050A; Semrock), mercury lamp (U-HGLGPS; Olympus), a CMOS camera (Zyla 4.2; Andor), and an optical table (RS-2000; Newport). Projection of the image to the camera was made at 130 nm per pixel with 2×2 binning. Sequential images of cells were acquired by the imaging software (Solis; Andor) as 16-bit images with a CMOS camera under 1-s intervals for immobilized cells under epi-illumination, and converted into a sequential TIF file without any compression.

For aerotaxis assay without flagella labeling, the cells were visualized under dark-field microscope (IX83; Olympus) equipped with 10× objective lens (UPLAPO 10×, 0.4 N.A.; Olympus), dark-field condenser (U-DCD; Olympus), halogen lamp (IX-HLSH100; Olympus), a CMOS camera (Zyla 4.2; Andor), and an an optical table (RS-2000; Newport). Projection of the image to the camera was made at 0.65 μm per pixel. Sequential images of cells were acquired by the imaging software (Solis; Andor) as 16-bit images with a CMOS camera under 1-s intervals for free-swimming, and converted into a sequential TIF file without any compression.

For competition assays, labeled cells were visualized under a fluorescence microscope (IX83; Olympus) equipped with 10× objective lens (UPLAPO 10×, 0.4 N.A.; Olympus), a filter set (59022; Chroma), mercury lamp (U-HGLGPS; Olympus), dual-view imaging system (FF560-FDi01, FF03-535/50 and BLP01-568R; Semrock), a CMOS camera (Zyla 4.2; Andor), and an optical table (RS-2000; Newport). Projection of the image to the camera was made at 1.3 μm per pixel with 2×2 binning. Sequential images of cells were acquired by the imaging software (Solis; Andor) as 16-bit images with a CMOS camera under 1-s intervals for immobilized cell under epi-illumination, and converted into a sequential TIF file without any compression. All data were analyzed by ImageJ 1.48v (http://rsb.info.nih.gov/ij/) and its plugins.

## Flow experiments

The flow chamber was assembled by taping a coverslip with glass slide, as previously described [49]. Inlet and outlet ports were made by boring through the slide glass with a high-speed precision drill press equipped with a diamond-tipped bit [2.35 mm diameter (ICHINEN MTM)]. A sample chamber was prepared from a glass slide and a coverslip (Matsunami glass) and double-sided tape ($\sim$100 μm thick, 3M). After the assembly, the flow chamber was incubated for 60 minutes at 120˚C to ensure tight adhesion of the slide and coverslip. Inlet and outlet ports (IDEX Health & Science) were attached with hot-melt adhesive (Goot; Taiyo Electronic IND). The total volume of the sample chambers was $\sim$10 μL. A syringe pump (Legato 210P; Kd Scientific) was used to control the flow rate of the buffer.

Labeled cells were suspended in PBS buffer in the presence of 50 μM CCCP (Sigma-Aldrich). The cell suspension was pipetted into the chamber and left to sit for 10 minutes to allow for binding of cells to the glass surface. The flow rate was calculated to be 4.6 mm/s by measuring the velocity of unattached, free-flowing cells in the sample chamber.

## Aerotaxis competition assays

A sample chamber was assembled by taping coverslips with double-sided tape (~150 μm thick, NW-K; Nichiban). WT and Δ*pgp1* cells were labeled with DyLight 594 and 488, respectively. After labeling, each cell suspension was adjusted to O.D.$_{600}$ 1.0 in MH broth + 0.5% methylcellulose. Equal volumes of each cell suspension were then mixed thoroughly in the same tube and pipetted into the sample chamber.

The microscope objective was focused at the tape/sample-chamber interface and recording was started immediately after injection of the cell mixture at a rate of 1 frame/second.

## Electron microscopy for measurement of flagellar filament lengths

For measurement of flagellar filament lengths, strains of interest were grown overnight on MHT agar and suspended in PBS to an O.D.$_{600}$ of ~1.5. Cell suspensions were then mixed 1:1 with 2% glutaraldehyde in $H_2O$.

3 μL of cell suspensions were then applied to glow-discharged Formvar-carbon copper grids (Agar scientific) and negative stained with 1% phosphotungstic acid (pH 7.5).

Micrographs were captured at 4,400x magnification on a FEI Tecnai T12 Spirit 120 kV electron microscope. Filament lengths were measured by hand using the freehand selection tool in the ImageJ FIJI software suite.

## Quantification and statistical analysis

Measurement of single-cell velocities were done by hand in FIJI (ImageJ) and values entered into the Graphpad Prism statistical anaylsis software suite (version 8.2.0). The average velocity are indicated as red bars in figures, with error bars representing the SEM. For comparison of two sets of velocity measurements and determination of significance, an unpaired t test was performed. For comparison of >2 groups, an ordinary one-way ANOVA test was performed.

## Supporting information

**S1 Fig. The *flaA*^S397C allele does not impact swimming ability.** Motility plates with or without methylcellulose added were inoculated with either DRH212 or EJC28 and allowed to incubate for ~24 hours. In both low-viscosity and high-viscosity motility agar, EJC28 were found to swim as well as DRH212 (**A** and **B**). Swim halo diameters in **B** are the average of five replicates, with error bars representing the SEM. To determine whether swimming velocity was impacted in EJC28 relative to DRH212, and whether labeling with DyLight and FM 4–64 had an effect on swimming, cultures of DRH212 and labeled cells were observed with 20x objective lens phase contrast microscopy. Both DRH212 and labeled EJC28 were found to swim at comparable velocities (**C**), demonstrating that neither the *flaA*^S397C mutation itself nor fluorescent labeling impacts swimming ability.
(TIFF)

**S2 Fig. Deletion of *fliI* reduces the number of flagella per cell, but does not impact flagellar length.** Cells of EJC28 and Δ*fliI* (EJC30) were applied to formvar-coated TEM grids, negative stained with phosphotungstic acid and observed at 4,400x magnification (**A**). While the Δ*fliI* mutant constructed fewer flagella/cell, flagellar length was comparable to WT (**B**). The flagellar filaments of EJC30 were observed to have an unlabeled FlaB portion similar in length to the WT (**C**), and the fluorescent FlaA filament for both WT and EJC30 were found to be approximately the same (**D**), indicating that the composition of the flagellar filament of the Δ*fliI* mutant is similar to WT. For **B** and **D**, filament length was determined by measuring by hand in ImageJ (FIJI) and plotted in Graphpad Prism, with error bars representing the SEM. For **B**, 45 WT filaments and 30 Δ*fliI* filaments were measured. For **D**, 31 WT filaments and 30 Δ*fliI* filaments were measured.
(TIFF)

**S3 Fig. Singly-flagellated Δ*pgp1* cells are slower than doubly-flagellated Δ*pgp1* cells.** *fliI* was deleted in the Δ*pgp1* straight-cell background in order to determine how much a helical cell body shape contributes to propulsion in high viscosity media. Similar to the Δ*fliI pgp1*^+ helical strain (**Fig 1F** and **S4 Movie**), singly-flagellated straight cells were found to be slower than doubly-flagellated straight cells, with the singly-flagellated wrapped cells being the slowest of the three.
(TIFF)

**S4 Fig. A phylogenetic tree of selected flagellins indicating that Campylobacter and Shewanella paralogous duplications of flagellins were independent.** Sequences of the conserved N- and C-terminal flagellin regions were aligned using FSA; unconserved gaps were removed using T-coffee; and the phylogeny was determined using RAxML and visualized with FigTree.
(TIFF)

**S5 Fig. Schematic of *fla* alleles generated for this study.** With the exception of strain WPK440 (**S3 Movie**), all *fla* cysteine alleles generated for this study were chromosomally encoded at the native *flaAB* locus. The WT strain for this study, EJC28 (*flaA*^S397C^), has a WT copy of *flaB* expressed from its native σ^54^ promoter (**A**). Our original *flaB* cysteine allele, *flaB*^S398C^, (**B** and **C**, **S16** and **S17 Movies**) was found to label poorly relative to the S397C allele. Consequently, for flow chamber experiments (**Fig 5**, **S18** and **S19 Movies**), we generated a single-flagellin mutants, harboring either *flaA*^S397C^ or *flaB*^S397C^ at the *fla* locus. In each case, the flagellin is expressed from the *flaA* σ^28^ promoter (**D** and **E**).
(TIFF)

**S6 Fig. Cells in the middle of the sample chamber swim slower than those at the edges.** When cells were tracked using 20x magnification phase-contrast microscopy (no fluorescent labeling), cells that were in the middle of the sample chamber swam at approximately half the velocity of cells near the taped edges of the sample chamber. This is presumed to be due to lower oxygen concentration in the middle of the sample chamber compared to near the porous, double-sided tape used to construct sample chambers, leading to a reduced proton motive force (PMF) to drive flagellar motor rotation.
(TIFF)

**S7 Fig. Deletion of *pgp1* impacts swimming speed and penetrance of high viscosity motility agar.** In regular motility agar (MH + 0.4% agar) the Δ*pgp1* mutant was found to swim nearly as well as WT, as judged by the diameter of the swim halo (2.88 cm vs. 3.70 cm, respectively. Values are mean of 5 replicates for each with error bars representing the SEM). In high-viscosity motility agar (MH + 0.4% agar + 0.3% methylcellulose (MC)), however, the Δ*pgp1* mutant was found to be incapable of penetrating and swimming through the agar. Rather, the straight cell mutant spread across the surface of the media (**A** and **B**). Using low magnification (20x) phase contrast microscopy, Δ*pgp1* cells in MH + 0.5% MC were found to swim at ~50% the velocity of WT cells, as has been previously reported.
(TIFF)

**S8 Fig. All-FlaA and all-FlaB are impaired for swimming through complex environments relative to WT.** In both regular and high-viscosity motility agar, the all-FlaA and all-FlaB mutants were found to swim with comparable efficiency, but both are inferior to WT with its composite filament assembled from both flagellin types (**A** and **B**). Values in **B** are the average of 5 replicates for each strain and condition, with error bars representing the SEM.
(TIFF)

**S1 Movie. The *C. jejuni* motor rotates at ~100 Hz.** Video captured at 1600 frames/second revealed that *C. jejuni*'s flagella rotate at ~100 revolutions/second. Area, 8.1 μm × 6.0 μm for 0.14 s.
(AVI)

**S2 Movie. *Campylobacter jejuni* wraps its leading flagellar filament around the cell body.** When fluorescently-labeled cells of EJC28 were observed swimming in MH broth, approximately 50% were found to wrap their leading filament around the cell body during swimming. When the swimming medium was changed to MH + ≥0.3% MC, almost all cells were wrapped. Area, 31.2 μm × 26.0 μm for 2.75 s.
(AVI)

**S3 Movie. The leading, wrapped flagellum is actively rotating.** Labeled WPK440 (Δ*flaAB*::*cat-rpsL* pRY108::*flaA*^S398C^) were suspended in MH + 25% Ficoll 400, which slows down cell

body rotation relative to filament rotation. We observed that the filament rotated around the arrested cell body, demonstrating that the wrapped flagellum is actively rotating. Area, 10.4 μm × 7.2 μm for 2.64 s.
(AVI)

**S4 Movie. The wrapped flagellum contributes to cell propulsion during swimming.** Singly-flagellated, Δ*fliI* cells with wrapped filaments are capable of swimming, albeit more slowly than either singly-flagellated unwrapped cells and doubly-flagellated WT cells. Area, 23.4 μm × 19.6 μm for 0.55 s.
(AVI)

**S5 Movie. Changing swimming direction involves a change in wrapped-filament polarity.** By fluorescently labeling EJC28, we were able to observe filament behavior during directional switching events. During a switch in swimming direction, the wrapped leading filament unwraps from the cell body to become the unwrapped lagging filament, allowing the previously unwrapped lagging filament to wrap around the cell and become the leading filament. Area, 13.0 μm × 7.2 μm for 0.33 s.
(AVI)

**S6 Movie. The *C. jejuni* flagellar filament maintains a LH helix during directional switching.** Although the FlaA portion of the filament exhibits considerable flexibility, we were unable to identify any point in the switching process, i.e. wrapping and unwrapping, when the filament switched handedness to a RH helix. Area, 9.9 μm × 8.6 μm for 0.41 s.
(AVI)

**S7 Movie. The cell body is a right-handed helix, and the flagellar filament is a left-handed helix.** Using TIRFM, the flagellar filament and cell body were found to have opposite handedness. Counter-rotation of the RH cell body and the LH filaments produce coherent, cooperative thrust. Area, 23.4 μm × 17.6 μm for 0.27 s.
(AVI)

**S8 Movie. A Δ*cheY* mutant still exhibits wrapped-mode swimming.** Wrapped cells were still observed in a Δ*cheY* background, indicating that the motors of both the wrapped and unwrapped flagella rotate CCW. The presence of wrapped cells in this background also suggest that wrapping of the leading filament does not require CCW-to-CW switching. Area, 15.6 μm × 15.6 μm for 3.00 s.
(AVI)

**S9 Movie. Paralyzed flagella do not wrap.** We would occasionally observe cells with one paralyzed flagellum. While the cause of filament paralysis is unclear, these filaments do not wrap around the cell. This indicates that wrapping is an active process requiring a rotating flagellum. Area, 16.0 μm × 10.1 μm for 0.60 s.
(AVI)

**S10 Movie. The inability of straight cells to unwrap results in a run-pause-run swimming style.** WT, helical *C. jejuni* exhibits darting motility, characterized by rapid, repetitive runs and reversals as it chemotaxes toward a preferred site in its environment. In the Δ*pgp1* background, the inability to unwrap upon CCW-to-CW motor switching results in a run-pause-run swimming style, as opposed to run-reverse-run. Area, 8.1 μm × 15.6 μm for 0.84 s.
(AVI)

**S11 Movie. The inability of straight cells to unwrap results in an abnormal run-reverse-run swimming style.** In addition to the run-pause-run phenotype, Δ*pgp1* cells exhibited run-

reverse-run swimming, but without unwrapping of the leading filament. As a consequence of being unable to unwrap, the unwrapped filament switches between being the lagging and leading filament, depending on whether the motor is running CCW or CW, respectively. As CW rotation typically only occurs for short periods of time, and since CW rotation is slower, these cells do not make long runs upon reversal as WT does. The functional consequence of this type of run-reverse-run in a non-unwrapping cell is a failure to make a significant change in swimming trajectory, unlike in cells that do unwrap. Area, 12.5 μm × 8.6 μm for 1.03 s. (AVI)

**S12 Movie. Straight cells are outcompeted by helical cells in an aerotaxis competition.** By independently labeling straight cells and WT cells with spectrally non-overlapping fluorophores (DyLight 488 and 594), we were able to observe the population-level tactic behavior each in direct competition with one another. The top panel in movie 8 is the Δ*pgp1* mutant labeled with DyLight 488, the bottom panel is WT labeled with DyLight 594. The WT swarm reached the taped border of the sample chamber faster than the Δ*pgp1* swarm. Additionally, the Δ*pgp1* swarm is both less dense as it sweeps toward the chamber border, leaving behind many stragglers, and accumulates to a significantly lower density at the border than the WT. Video was captured at 1 frame/second. Area, 1100 μm × 850 μm for 120 s. (AVI)

**S13 Movie. WT cells aerotax to the edge of the sample chamber and form a high-density swarm there.** Prior to the dual-labeling competition assays (**Fig 4** and S12 Movie), we observed unlabeled WT and Δ*pgp1* cells individually by low-magnification dark-field microscopy. Although there was variability from slide-to-slide (hence the decision to compete the two strains directly in the same sample chamber), the WT (**S13 Movie**) consistently reached the sample chamber border faster, left fewer stragglers and formed a tighter, denser swarm at the border than the Δ*pgp1* mutant (**S14 Movie**). Videos were captured at 1 frame/second. Area, 1250 μm × 780 μm for 500 s. (AVI)

**S14 Movie. WT cells aerotax to the edge of the sample chamber and form a high-density swarm there.** Prior to the dual-labeling competition assays (**Fig 4** and **S12 Movie**), we observed unlabeled WT and Δ*pgp1* cells individually by low-magnification dark-field microscopy. Although there was variability from slide-to-slide (hence the decision to compete the two strains directly in the same sample chamber), the WT (**S13 Movie**) consistently reached the sample chamber border faster, left fewer stragglers and formed a tighter, denser swarm at the border than the Δ*pgp1* mutant (**S14 Movie**). Videos were captured at 1 frame/second. Area, 1250 μm × 780 μm for 500 s. (AVI)

**S15 Movie. The Δ*cheY* mutant does not aerotax.** As a control, the *cheY* mutant was checked for aerotactic behavior. As expected, this mutant did not migrate to the sample chamber border. A video of the WT (upper panel) is provided for comparison. Area, 845 μm × 913 μm for 350 s. (AVI)

**S16 Movie. Filaments with FlaB distributed along their length are impaired for wrapping.** The majority of labeled cells in a *flaA*⁺ *flaB*^S398C background have short, cell-proximal fluorescent stubs, consistent with FlaB being localized to the base of the flagellar filament. Occasionally, however, cells with FlaB distributed along their length were observed. Many of these cells did not wrap their filaments when swimming in high-viscosity media, suggesting that the presence of FlaB throughout the filament increases filament rigidity. Area, 16.6 μm × 32.9 μm for

3.00 s.
(AVI)

**S17 Movie. An all-FlaB mutant is impaired for wrapping when swimming at lower velocity.**
After noticing that filaments with FlaB distributed along their length (in a *flaA*$^+$ *flaB*$^{S398C}$ background) tended not to wrap, we constructed an all-FlaB$^{S398C}$ mutant for observation. All-FlaB$^{S398C}$ cells were found to wrap their leading filaments when observed near the border of the sample chamber, where swimming velocity is higher. Slower swimming cells in the middle of the same sample chamber were found to be predominantly unwrapped. Area, 31.2 μm × 26.0 μm for 1.25 s.
(AVI)

**S18 Movie. All-FlaA filaments stretch more under flow than all-FlaB filaments.** By labeling paralyzed all-FlaA$^{S397C}$ and all-FlaB$^{S397C}$ cells and allowing the filaments to adhere to the glass cover slip of a flow chamber, we were able to compare the stretching ability of each filament type. The all-FlaA$^{S397C}$ filament (**S18 Movie**) was found to stretch up to ~30%, while the all-FlaB$^{S397C}$ filament (**S19 Movie**) only stretched ~10% under the same flow conditions. Movie 10: Area, 10.8 μm × 4.2 μm for 43 s. Movie 11: Area, 17.9 μm × 6.9 μm for 67 s.
(AVI)

**S19 Movie. All-FlaA filaments stretch more under flow than all-FlaB filaments.** By labeling paralyzed all-FlaA$^{S397C}$ and all-FlaB$^{S397C}$ cells and allowing the filaments to adhere to the glass cover slip of a flow chamber, we were able to compare the stretching ability of each filament type. The all-FlaA$^{S397C}$ filament (**S18 Movie**) was found to stretch up to ~30%, while the all-FlaB$^{S397C}$ filament (**S19 Movie**) only stretched ~10% under the same flow conditions. Movie 10: Area, 10.8 μm × 4.2 μm for 43 s. Movie 11: Area, 17.9 μm × 6.9 μm for 67 s.
(AVI)

**S20 Movie. The all-FlaA mutant is more likely to have straight filaments.** A number of cells in the all-FlaA$^{S397C}$ background were observed with straight, rotating filaments. This was not observed in the all-FlaB strains and was rare in the WT background. This supports the notion that the FlaB portion of the filament serves to promote helicity in the FlaA portion of the filaments. Area, 13.0 μm × 10.4 μm for 0.95 s.
(AVI)

**S21 Movie. Unwrapped and wrapped Δ*cheY* cells swim at approximately the same velocity in low viscosity.** When the Δ*cheY* mutant was observed in MH broth lacking any methylcellulose, both uwrapped and wrapped cells were seen, similar to WT. Unwrapped and wrapped Δ*cheY* cells both swim at low velocity. This suggests that differences between the leading and lagging filament (*e.g.* length or helical angle) allow a cell with unwrapped, opposing flagella to swim, rather than asymmetric rotation of the two motors. Area, 15.6 μm × 19.5 μm for 2 s.
(AVI)

## Acknowledgments

The authors thank Paul Simpson for assistance with electron microscopy and Seiji Iwata for technical assistance.

   Lead contact and materials availability.

   All strains generated for this study are available upon request. Further information and requests for resources and reagents should be directed to and will be fulfilled by the Lead Contact, Morgan Beeby (m.beeby@imperial.ac.uk).

## Author Contributions

**Conceptualization:** Eli J. Cohen, Daisuke Nakane, David R. Hendrixson, Takayuki Nishizaka, Morgan Beeby.

**Formal analysis:** Eli J. Cohen, Daisuke Nakane.

**Funding acquisition:** Daisuke Nakane, David R. Hendrixson, Takayuki Nishizaka, Morgan Beeby.

**Investigation:** Eli J. Cohen, Daisuke Nakane.

**Methodology:** Eli J. Cohen, Daisuke Nakane, Yoshiki Kabata.

**Resources:** David R. Hendrixson, Takayuki Nishizaka, Morgan Beeby.

**Writing – original draft:** Eli J. Cohen, Morgan Beeby.

**Writing – review & editing:** Eli J. Cohen, Daisuke Nakane, David R. Hendrixson, Takayuki Nishizaka, Morgan Beeby.

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
