## [Decision Letter · Decision Letter 0]

10 Apr 2020

Dear Dr Beeby,

Thank you very much for submitting your manuscript "Campylobacter jejuni motility integrates specialized cell shape, flagellar filament, and motor, to coordinate action of its opposed flagella" for consideration at PLOS Pathogens. As with all papers reviewed by the journal, your manuscript was reviewed by members of the editorial board and by several independent reviewers. The reviewers appreciated the attention to an important topic. Based on the reviews, we are likely to accept this manuscript for publication, providing that you modify the manuscript according to the review recommendations.

All three reviewers feel enthusiastic about the study and think that it provides valuable input to the biophysical aspect of bipolar flagella-mediated motility. They have some questions and suggestions that the authors should respond to and reflect on the manuscript.

Sincerely,

Tomoko Kubori, Ph.D.

Associate Editor

PLOS Pathogens

Nina Salama

Section Editor

PLOS Pathogens

Kasturi Haldar

Editor-in-Chief

PLOS Pathogens

orcid.org/0000-0001-5065-158X

Michael Malim

Editor-in-Chief

PLOS Pathogens

orcid.org/0000-0002-7699-2064

All three reviewers feel enthusiastic about the study and think that it provides valuable input to the biophysical aspect of bipolar flagella-mediated motility. They have some questions and suggestions that the authors should respond to and reflect on the manuscript.

Reviewer Comments (if any, and for reference):

Reviewer's Responses to Questions

**Part I - Summary**

Reviewer #1: A complicated movement of bipolar flagella of C.jejuni was analysed using Total Internal Reflection Fluorescence Microscopy.

Visualysing each flagellum and the cell body by fluorescent labelling clearly showed the shape and handedness of flagella on the cell body.

Flagellum on the head wrapped the cell body. Handedness of flagella and the cellbody explained how they could produce a strong force in viscous media without tangling each other.

Results were well written to describe the phenomenon. An excellent work worth publishing in this jouranal. However I have several questions before publication.

Reviewer #2: This is a really interesting manuscript that attempts to solve the old problem of how bacteria with bipolar flagella coordinate these rotary motors to achieve synchronous motion. The authors discover that in high viscosity environments the leading flagellum wraps up around the cell body, thereby flipping itself to be coordinated with the lagging flagellum. The authors argue that the leading flagellum and the helical geometry of the cell body itself both contribute to the propulsion. This is a new model for bipolar helical motility and therefore exciting.

Reviewer #3: Campylobacter jejuni is amphitrichously flagellated and swims using two flagella located at each cell pole. Remarkably, these bacteria reach swimming speeds of up to 100 microns per second. How C. jejuni coordinates rotation of two opposed flagellar motors and how two flagella contribute to the motility behavior of the bacteria remains poorly understood. In the present manuscript, Cohen and colleagues used high-speed fluorescent microscopy to visualize the flagellar filament and analyze the swimming behavior of C. jejuni. Using a clever combination of various C. jejuni mutants and sophisticated microscopy approaches, the authors found that C. jejuni wraps the leading flagellar filament around the cell body, while both leading and lagging filaments rotate counter-clockwise. These results allowed the authors to propose a novel model for C. jejuni motility, where the wrapped-mode swimming behavior enables C. jejuni to efficiently move through high-viscosity environments, such as the mucus environment of the gastrointestinal tract.

The manuscript has been a pleasure to read, the data is well-presented and supports the author’s conclusions. I especially appreciated the use of a DcheY mutant to unequivocally demonstrate that both filaments of C. jejuni rotate counter-clockwise by default.

**Part II – Major Issues: Key Experiments Required for Acceptance**

Reviewer #1: 1. Although I have completely understood usage of "wrapped" and "unwrapped" in the text, I felt uneasy with usage in some sentences.

For example, 163: filament is wrapping the cell body but not wrapped.160: wrapped filament.

2. 172. It is better to use the pitch distance and the diameter of a helix than pitch angle, which is often used in engineering and particle physics.

Reviewer #2: 1. A key question is whether the leading flagellum provides propulsive force. I do not understand the authors' line of reasoning for why they conclude that it does. Cells with wrapped flagella move faster, but couldn't this be an inverse causation: what if high speed causes wrapping?

2. The authors use several mutants to try to address questions like flagellar number of cell shape, but they assume that these mutants do not disrupt other properties of the cell. For example, the straight mutant affects cell wall - what if it affects cell stiffness, which would also affect speed? Or if the flagella of fliI mutants are weaker?

Reviewer #3: None.

**Part III – Minor Issues: Editorial and Data Presentation Modifications**

Reviewer #1: 126: what does "stored" mean?

386: E.coli and Salmonella should be in Italic.

423: "yank" may be "pull".

448: typo, facilate.

Reviewer #2: I really enjoyed this paper and think it would be a valuable contribution to understanding the biophysical basis of bacterial motility. However, I wonder if its best home is really PLoS Pathogens. If the editor and authors think yes, then I am highly supportive. I just hope it will reach the right audience, as opposed to something like Biophysical Journal.

Reviewer #3: 1) Abstract, line 19: Enterobacteriaceael? Perhaps rephrase to ‘the norm of Enterobacteriaceae’?

2) Single flagellated cells. The authors used a DfliI mutant, which assembles flagella less frequently and thus most cells have none or only one flagellum. While Fig. S2 shows that assembled flagella have approximately the same length, can the authors exclude that the ratio of the two flagellins (FlaA/FlaB) is not changed, which would result in different properties of the filament? This might also account for the observed decreased motility?

Are there any cells in the WT, which only produce one flagellum? Even if there are only a few, it might be useful to compare motility of these uniflagellated cells with uniflagellated cells of the DfliI mutant.

3) The DcheY mutant experiment to demonstrate that the default rotational direction of the motor is CCW is a beautiful example of how simple genetics can answer a fundamental problem.

4) Does flagellin glycosylation contribute to the observed cell body wrapping of the filament? Is flagellin in the flaA_S397C mutant still glycosylated?

5) Concerning the question of how unwrapped cells are able to swim in low-viscosity media (i.e. both flagella at the opposing cell poles are unwrapped and would cancel each other’s propulsion). One possibility is that – contrary to the proposed model – the motors at the opposing cell poles rotate in different directions. Did the authors analyze the swimming behavior of their DcheY mutant in low-viscosity media? This would at least ensure that both flagella are rotating CCW and might support alternative hypotheses, e.g. that different filament lengths or helical pitches enable swimming of unwrapped cells.

6) Line 384 + 396: define the abbreviations ‘LH’ and ‘RH’ in the manuscript text.

7) Line 386: E. coli and Salmonella should be italics

8) Fig. 1B y-axis should be labeled (e.g. ‘proportion of cells’)

9) Fig. 2: The rotational direction of flagella and the cell body is indicated by arrows. Consider to use different colors for the arrows indicating rotational direction of flagella and the cell body. Also, for the first schematic, both the top flagellum and cell body are indicated to rotate CCW. This is wrong for the cell body, which should rotate CW?

10) Fig. 3: Bad quality of the schematic and figure labels

11) Fig. 4A: Scale bar is missing.

12) Fig. 4B, D, E: Should read ‘∆pgp1’

13) Fig. 5: Bad quality of the schematic and figure labels

14) Fig. 1B, 4E, 5C, S1B, S2B, S7B, S8B: Information concerning the number of replicates or samples, as well as error bars are missing and should be indicated in the figure legend.

15) Fig. S2A+B: Should read ‘∆fliI’.

16) Fig. S2B + S2: y-axis label should read ‘µm’

17) Fig. S7: Bad quality of the figure labels

PLOS authors have the option to publish the peer review history of their article (what does this mean?). If published, this will include your full peer review and any attached files.

Reviewer #1: No

Reviewer #2: No

Reviewer #3: No
---

## [Editor Report · Decision Letter 1]

1 May 2020

Dear Dr Beeby,

Thank you very much for submitting your manuscript "Campylobacter jejuni motility integrates specialized cell shape, flagellar filament, and motor, to coordinate action of its opposed flagella" for consideration at PLOS Pathogens. 

I am grateful that the authors comprehensively addressed the comments raised by all the reviewers. Before I will officially accept the manuscript, I would like to request the authors to supply the video of the delta cheY in low viscosity media according to the Reviewer 3's Minor comment 5), as I think that the information is important for supporting the model. Please give the minor change of the text accordingly.

Sincerely,

Tomoko Kubori, Ph.D.

Associate Editor

PLOS Pathogens

Nina Salama

Section Editor

PLOS Pathogens

Kasturi Haldar

Editor-in-Chief

PLOS Pathogens

orcid.org/0000-0001-5065-158X

Michael Malim

Editor-in-Chief

PLOS Pathogens

orcid.org/0000-0002-7699-2064

I am grateful that the authors comprehensively addressed the comments raised by all the reviewers. Before I will officially accept the manuscript, I would like to request the authors to supply the video of the delta cheY in low viscosity media according to the Reviewer 3's Minor comment 5), as I think that the information is important for supporting the model. Please give the minor change of the text accordingly.
---

## [Editor Report · Decision Letter 2]

11 May 2020

Dear Dr Beeby,

We are pleased to inform you that your manuscript 'Campylobacter jejuni motility integrates specialized cell shape, flagellar filament, and motor, to coordinate action of its opposed flagella' has been provisionally accepted for publication in PLOS Pathogens.

Best regards,

Tomoko Kubori, Ph.D.

Associate Editor

PLOS Pathogens

Nina Salama

Section Editor

PLOS Pathogens

Kasturi Haldar

Editor-in-Chief

PLOS Pathogens

orcid.org/0000-0001-5065-158X

Michael Malim

Editor-in-Chief

PLOS Pathogens

orcid.org/0000-0002-7699-2064
---

## [Editor Report · Acceptance letter]

29 May 2020

Dear Dr Beeby,

We are delighted to inform you that your manuscript, "Campylobacter jejuni motility integrates specialized cell shape, flagellar filament, and motor, to coordinate action of its opposed flagella," has been formally accepted for publication in PLOS Pathogens.

Best regards,

Kasturi Haldar

Editor-in-Chief

PLOS Pathogens

orcid.org/0000-0001-5065-158X

Michael Malim

Editor-in-Chief

PLOS Pathogens

orcid.org/0000-0002-7699-2064